# A Review of the Production, Recycling and Management of Marine Plastic Pollution

**Ibrahim Issifu *** **and U. Rashid Sumaila ***

Fisheries Economics Research Unit, Institute for the Oceans and Fisheries, University of British Columbia, Vancouver, BC V6T 1Z4, Canada

**\*** Correspondence: i.issifu@oceans.ubc.ca (I.I.); r.sumaila@oceans.ubc.ca (U.R.S.)

**Abstract:** The human attachment to plastic has intensified recently due to its lightweight, versatility, low-cost and durability and so has the damage to the marine environment as marine plastic pollution has correspondingly increased. As a result, there has been increasing concern on the issue of marine plastic pollution. Policy-based organizations such as the United Nations Environment Programme have drawn public attention to the scope, magnitude and impacts of marine pollution in recent decades. Research on marine pollution can play a significant role in contributing to policy-making processes in support of the United Nations Sustainable Development Goal on Life Below Water (SDG 14), by providing scientific analysis on the effects and sources of marine plastic pollution. This paper provides a theoretical and empirical overview of marine plastic pollution and its potential effects on marine ecosystems. It also discusses SDGs that are relevant to marine plastic pollution and suggest priorities for further research.

**Keywords:** marine plastic pollution; sustainable development goals

## 1. Introduction

Plastic has become ubiquitous in human society, as of 2015, about 8.5 billion metric tons of plastic was estimated to have been produced globally since the first production of synthetic plastic in the early 20th century [1]. With an annual growth of 4% between 2010 and 2015, more plastic has been produced in the last two decades compared to the previous 50 years [2]. Cheap production of primary plastic packaging materials from petroleum products has led to the creation of the biggest virgin plastic market compared to recycled plastics resulting in substantial amount of plastic in landfills and the natural environment [3]. In 2010, close to 12.7 million tons of mismanaged land-based plastic waste was estimated to have entered the oceans [4]. The situation is acute for many middle-income countries who have become the epicenters for plastic leakage. The current review seeks to address the following: What has been the outcomes of selected studies on the effect of plastic pollution on marine ecosystems? Are there studies pointing to significant reduction in the levels of plastic pollutants? The purpose of this paper is to present both theoretical and empirical overview of marine plastic pollution based on existing studies and suggest future recommendations in accordance with the lessons learnt from the current study.

By the end of 2015, 16 of the top 20 nations contributing to marine plastic pollution (MPP) were middle-income economies, whose rapid economic growth outpace waste management infrastructure expansion [4]. Plastic pollution seems to be a single greatest threat to marine health due to its abundance and longevity once leaked into the marine environment. Should global demand for and uncontrolled disposal of plastic waste go unchanged, it is suggested that there will be more plastic in the ocean than fish biomass by 2050 [5].

Although the discovery of plastic remains a novelty without well-thought out management of its waste, the global environment stands to suffer its negative effect. Given the closely intertwined nature of marine plastic pollution, the call for its careful management and prevention is strategic and timely [6].

To ameliorate the effect of marine pollution, significant initiatives targeted at reducing dumping of plastic waste into oceans has been carried over the past three decades. The International Convention for the Prevention of Pollution from Ships (MARPOL) began publishing a series of articles to strengthen its disciplines on marine pollution in 1978. Since its introduction about 140 states have approved MARPOL, covering 97.5% of global shipping tonnage [7]. The MARPOL Directive, as amended in 2012, focused on stopping pollution from vessels. As stated by the International Maritime Organization, MARPOL forbids dumping of plastic anywhere in the ocean, while the discharge of other waste is severely restricted to coastal waters and "Special Areas" (i.e., areas designated special due to sea traffic and ecological conditions).

The United Nations Convention on the Law of the Sea (UNCLOS) has also hosted workshops and conferences to highlight the conservation and sustainable handling of marine resources via legal processes as contained in the UNCLOS' protocol entitled 'The Future We Want.' Furthermore, Articles 192–237 of UNCLOS outline basic rules to address marine pollution from various sources. However, the protocols of UNCLOS fail to specify the types of measures necessary, leaving it to individual governments to adopt local and regional laws and regulation to address sources of marine pollution. Recent developmental goals such as the United Nations (UN) Agenda 2030 for Sustainable Development, approved in 2015, provides an overarching plan to guide global actions. Out of the 17 Sustainable Development Goals (SDGs), three keys align directly and indirectly with reducing marine pollution and improving marine life (Goal 14, 12 and 3). Targets 14.1 and 14.2 specifically address the issue of marine pollution. Land-based causes of ocean pollution to be reduced significantly by 2025 are depicted in Table 1. To quickly achieve a reduction in MPP, short- and medium-term strategies must focus on campaigns to use plastic waste as inputs into other products (e.g., building and road construction materials) and work to recover and reprocess usable products that might otherwise be discarded (i.e., recycling programs). Long-term strategies should focus on sustainable initiatives such as reducing waste at source or production stage. The association and potential impact of the three SDGs (i.e., 3, 12 and 14) on MPP is illustrated by Figure 1. Here we provide a conceptual relationship of how achievements of three SDGs may result in reduction in the volume of plastics in oceans.

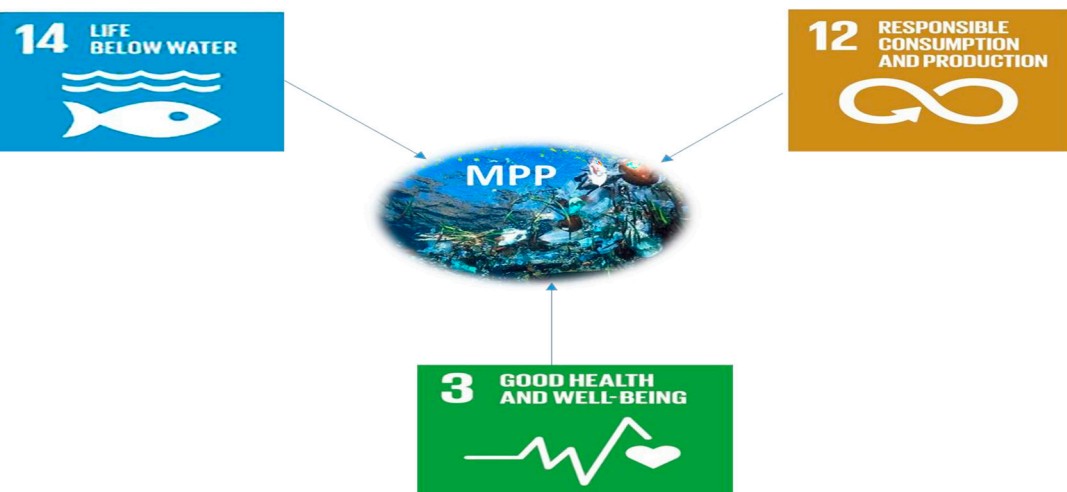

**Figure 1.** Potential impact of Sustainable Development Goals (SDGs) on marine plastic pollution. Presents a conceptual relationship of how achievements of three SDGs-SDG 3 (Good health and well-being); SDG 12 (Responsible consumption and production) and SDG 14 (Life below water) may reduce marine plastic pollution (MPP).

**Table 1.** Sustainable Development Goals targets related to marine litter. Sources: Adapted from [8].

| SDGs Target Index | SDG Targets Linked to Marine Debris |
|---|---|
| 6.3 | By 2030, the quantity of untreated wastewater should be halved. |
| 11.6. 1 | By 2030, minimize the adverse per capita environmental impact of cities, including by paying special attention to air quality and municipal and other waste management. |
| 12.1 | Implement the 10-year framework of programme on sustainable production and consumption of all countries acting, with advanced economies taking the lead, considering the development and capabilities of developing economies. |
| 12.2 | By 2030, achieve the sustainable management and efficient use of natural resources. |
| 12.4 | By 2020, achieve the environmentally sound management of chemicals and wastes throughout their life cycle. In accordance with agreed international frameworks and significantly reduce their release to air, water and soil in order to reduce their adverse impacts on human health and the environment. |
| 12.5 | By 2030, substantially minimize waste generation through prevention, reduction, recycling and reuse. |
| 12.b | Develop and implement tools to monitor sustainable development impacts for sustainable tourism that create jobs and promote local culture and products. |
| 14.1 | By 2025, prevent and significantly reduce marine pollution of all kinds, in particular from land-based sources, including marine debris and nutrient pollution. |
| 14.2 | By 2020, sustainably managed and protect marine and coastal ecosystems to avoid significant adverse impacts, including by strengthening their resilience and act for their restoration in order to achieve healthy and productive oceans. |
| 14.7 | By 2030, increase the economic benefits to Small Island Developing States and least developed economies from the sustainable use of marine resource, including through sustainable management of fisheries, aquaculture and tourism. |
| 14.a | Increase scientific knowledge, develop research capacity and transfer marine technology, considering the intergovernmental Oceanographic Commission Criteria and Guidelines on the Transfer of Marine technology, in order to improve ocean health and to enhance the contribution of marine biodiversity to the development of developing economies, in particular Small Island Developing States and least developed economies. |
| 14.c | Enhance the conservation and sustainable use of oceans and their resources by implementing international law as reflected in UNCLOS, which provides the legal framework for the conservation and sustainable use of oceans and their resources, as recalled in paragraph 158 of The Future We Want. |
| 15.5 | Take urgent and significant action to reduce the degradation of natural habitats, halt the loss of biodiversity and, by 2020, protect and prevent the extinction of threatened species. |

Notwithstanding these global efforts, research interest on the negative phenomena continue to rise due the exponential increase in the use of plastic in modern society [6].

Inadequate management of waste generated through plastic possess a significant threat to global efforts to improve life under water. The ultimate effects will be the significant reduction in economic welfare as a result of its impacts on marine ecosystem and ecological communities. This will also reduce revenues from marine-based tourism [9,10]. Studies on the effect of plastics has been conducted across the globe but these have remained scatted and, in most instances, computing the absolute amount of plastic discharged into the ocean is said to be difficult. The different environmental transport pathways further complicate the computational challenges associated with MPP [6].

## 2. Methods

The search for peer-review papers was conducted using search engines and citation databases such as Scopus, Google Scholar and Science Direct. To facilitated the search a combination of key words and phrases such as "marine pollution" "plastic pollution," "ocean pollutants," "environmental marine plastic," "sustainable development" were employed. Boolean operators such as AND, OR and NOT were also employed. In addition to the keywords and Boolean operators, the search was further narrowed down with the use of filters such as year of publication. The following inclusion criteria was adopted to help sift through the volumes of publications generated; relevance of paper to the topic based on stated keywords and the content of the abstract. We excluded student thesis and

non-academic documents from the review. A desktop review of each of the journal articles and content analysis was used to develop the respective themes and categorization of the articles. Special attention was given to the theoretical foundations that the selected papers were based on.

## 3. Categories of Academic Studies

We categorized the literature on MPP into three thematic themes: Exploratory empirical and theoretical. Exploratory studies can also be referred to as pioneering studies focus on the definitions of MPP [11]; identification of the sources and movement of plastic litter; and the estimation of the rate of plastic pollution to the marine environment. For instance, it has been estimated that 27.5% of all plastic litter ends up in the oceans [12]. Empirical studies aim to provide systematic evidence on the impact of MPP in the real world. However, they are limited by the lack of data on the amount of plastic entering the ocean [13].

The development of theoretical frameworks on MPP started with the Driver-Pressure-State Impact-Response (DPSIR) model, developed to explain how social and economic developments exert pressure on the environment leading to changes in the environment [14]. But the general conclusion appears to be that controlling MPP should concentrate on material reduction, increased Government investment to improve recycling capacity, increase the design of end-of-life recyclability, strategies to reduce littering and development of bio-based feedstocks [15].

### 3.1. Empirical Studies

In this section, we present and discuss the findings of some empirical studies on the sources and impacts of MPP.

### 3.1.1. MPP: Sources, Pathways and Estimation

Ingestion of plastics by marine organisms such as seabirds gained prominence as early as the 1960s [16]. The interactions between marine living organisms and persistent debris in the northwest Hawaiian Islands [17]. As the first systematic study on MPP reported in the 1960s, their findings provide evidence that the 100 seabirds that died had plastic debris in their abdomens at an average weight of 2 g of plastic per seabird after flushing their stomachs. They concluded that these indigestible plastics were ingested unintentionally at sea, suggesting that the huge size of some of these plastic litters might have led to the demise of seabirds. Added to this, significant interest in ingested plastic pellets hindering the digestive efficiency, sometimes resulting in deaths among seabirds were raised [17]. The subsequent findings of huge quantities of plastic items in the North Atlantic in the 1970s spurred research interest in MPP. Some novel studies reported the presence of plastics on the seabed, which affected a variety of marine organisms. The first comprehensive survey of marine plastics debris examined 100 Laysan albatross carcasses in the Southeast Islands in the Caribbean and North Atlantic and revealed that pellets of indigestible items were prevalent in the area but were concentrated close to mainland-based channels along the eastern seaboard in the US [18]. Unlike [17], they failed to find any pellets in marine living organisms sampled.

Notwithstanding these studies, global research agenda on marine plastic pollution gained momentum after the first two conferences organized by the US National Marine Fisheries Service in Honolulu between 1982 and 1984 [19]. The prevalence of litter in the open oceans is highlighted by numerous images of plastic showing in shorelines and flowing into rivers before entering the oceans and by the fact that every year large quantities of litter are collected by ocean cleanups around the world. For instance, the 2017 cleanup event showed that the dominated top items collected around the global based on item counts of coastal litter were all made of plastics, which was repeated during the 2018 cleanup event [20]. Meanwhile, a number of studies provide estimates of marine plastic debris [4,21]. As recently reported by Eriksen et al. [22], an estimated figure of the volume of plastics floating in the oceans, and found that more than five trillion pieces of plastic and approximately 268,940 tons are currently floating in the oceans. They estimation however excluded plastic litter on the seafloor.

The widespread use of single-use plastic and unmanaged disposal of litter along with pour waste management and recycling practices contribute to the growing accumulation of litter in the oceans. In terms of transportation pathways, leakages from municipal solid waste streams which ultimately end up in the seas have been viewed as an increasing source of plastic debris in the oceans [23].

Results from studies estimating MPP–covering both ocean and land-based sources–indicate these are very large [4,24,25]. The total plastic pollution of 15 million metric tons per year is estimated [12]. These estimates were based on compilation from previously published sources. The marine environment has become a substantial reservoir for plastic litter with huge negative effects [26]. As the definition of marine plastic debris deepened [22], necessitating studies on other sources and forms of marine debris [27,28]. New perspective to this debate found microplastic in Arctic polar waters and suggest that the accumulation of plastic can be attributed to transporting agents such as ocean currents, winds and tides [27]. These agents enhance the transport of plastic to remote regions far from the original sources. Inland populations contributed between 0.79–1.52 million tons per year of plastic to oceans through river transport [29]. Their findings were based on plastic inputs from inland areas (>50 km from the coastline) to oceans. By analyzing the distribution and abundance, plastic litter can be found in marine ecosystems, including beaches, shorelines, surface waters and on the seafloor [30].

### 3.1.2. Impact of MPP on Marine Ecosystems

The review of the selected peer-review papers shows significant academic interest on the effect of MPP on marine ecosystems. Feeding tests with plastic pellets had confirmed that small fish often ingested plastic pellets, which increase their mortality rates [31]. Floating plastic-litter such as fishing gear have been found to constitutes a navigation hazard, leading to death or injury of marine organisms, these often-clogged water intakes or interfering with ship propellers [26].

The early studies that respond to the growing awareness of floating plastic debris stranded on beaches, found that regular retention rate of containers and other distinct bottles varied across different beaches and that plastic bottles last longer on beaches than non-plastic bottles [32]. Active ocean currents culminated in low weekly retention rates (11%–29%) and spread litter throughout the Southern North Sea. Accordingly, employing manufacturers' identification symbols to examine the durability of containers, about 20% of plastic litter were made more than two years preceding stranding [32]. Data from remote Alaskan beaches was used to conduct one of the first extensive quantitative studies of beach debris and revealed that the amount of plastic debris increased from 2221 to 5367 items between 1972 and 1974 [33]. Plastic might be a cause of harmful compounds and other phthalates into marine ecosystems [33]. In this case, a study was undertaken in response to overreliance on qualitative analysis on the effect of plastic bottles on marine life by earlier researchers [33].

One of the most significant studies of the impacts of marine plastic litter on fisheries found that abandoned, lost and discarded fishing gear (ALDFG), otherwise well-known as ghost fishing (Ghost fishing occurs when traps or nets, discarded at sea, continue to cause significant mortality, often to already overexploited stocks.) [34] influence marine ecosystems and the life they support. Trammel nets and gill nets are the most significant causal factors in terms of gear lost and in terms of ghost fishing-related mortality. From their review of 76 publications and other sources of grey literature, over 5400 individuals from 40 different species were recorded as entangled in, or associated with, abandoned longlines [35]. It was predicted that removing or eliminating 9% of lost traps and nets would raise annual global landings by approximately 294,000 tons [36]. The global effect is that close to 52% of sea turtles may ingested plastic litter [13]. Considering the continuing detection of plastic entanglements by a growing cohort of marine animal species, marine environment has perhaps become a sizable sink for plastic waste across the globe. Figure 2 shows the share of species with records of entanglement in marine debris in 2015. For example, nine species of true seals have recorded about 47% marine litter entanglement.

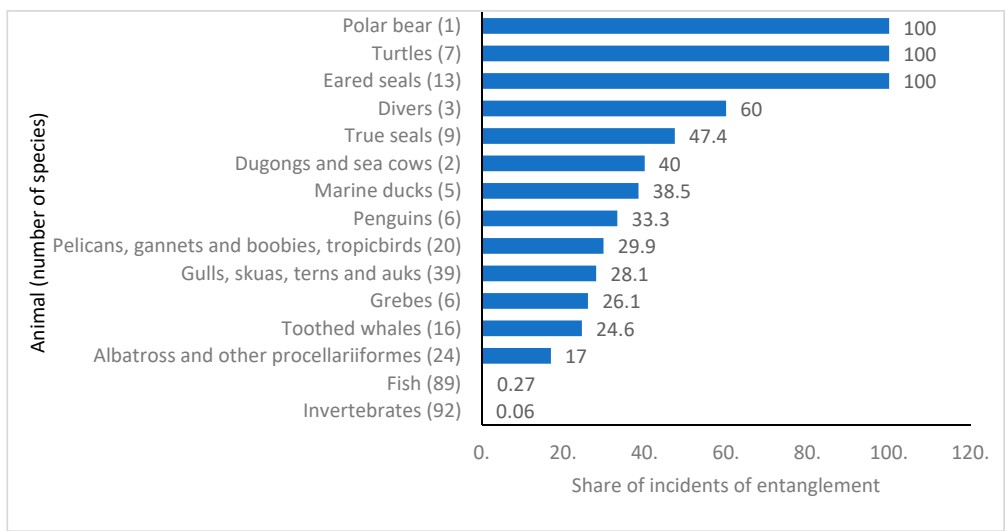

**Figure 2.** Share of incidents of entanglement. The number inside the parenthesis indicates number of species and the numbers after each blue bar denotes percent of the animals in each sample that were entangled. For example, seven species of turtles have recorded 100% marine litter entanglement; likewise, three species of divers recorded about 60% of marine debris entangled. [data source: Statista accessed online: https://www.statista.com/statistics/596960/share-of-species-with-with-records-of-entanglement-in-marine-debris/] (accessed on 1 March 2020).

The degradation of macroplastics into microplastics—those that directly leaked into the oceans as plastic particles less than 5 mm in diameter—are widely known as marine environment pollutants [8]. A mid-point estimation of 1.5 million tons of microplastics from the average of lower and upper ranges (0.8–2.5 million tons) of abrasion of car tyres, laundering of synthetic textiles, abrasion of road markings, fallout of city dust, release of marine coatings, spills of plastic pellets and microbeads from cosmetics entering oceans per year [37]. Synthetic textiles accounts for 35% of the world's sources of ocean microplastics while some 28% of the ocean's microplastics originated from car tires [37]. It is estimated that microplastics account for 94% of the Great Pacific Garbage Patch in the ocean [37].

The quantity of plastic waste that was mismanaged as well as the mass of plastic available to be transferred into the marine environment in the year 2010, an estimated 275 million tons of plastic waste was created by 192 coastal nations, with 4.8 to 12.7 million tons transported into the ocean [4]. Ten largest emitters of MPP worldwide, include China, Indonesia, Philippines, Vietnam, Sri Lanka, Thailand, Egypt, Malaysia, Nigeria and Bangladesh and it is predicted that by the year 2025, the cumulative amount of plastic waste feasible to get in the ocean will double without remarkable advances in waste collection and management strategy [4].

On the regional level, Asia accounts for 86% of plastic input share from rivers into oceans [29]. Current estimates show that only around 40% of plastic waste are collected on average across, Thailand, the Philippines, China, Vietnam and Indonesia [38]. One might argue that waste collection systems are insufficient in Asia and many developing countries, where the collection and disposal of domestic and imported waste are largely unregulated. Municipal solid waste (MSW) provides a mechanism for plastic debris as well as organic waste to be transported into the marine environment. Until recently, there has been no reliable evidence about the quantity of plastic in MSW. Data reported in [39] reveals that, in 2018, the share of plastics in global municipal solid waste was approximately 12%. The global flow of plastic resins as of 2016 and found that about 300 million metric tons of new plastic was produced and of this amount, 260 million metric tons were turned into waste and about 19% was unmanaged dump or leaks into oceans [3].

Building on the [4,8] simulated the distribution of plastic waste based on the predicted flow of plastic as a result of insufficient waste treatment and found a significant trans-boundary transport across

the Bay of Bengal. Likewise, global compilation of data on plastic litter in the water column across a broad range of rivers, found that plastic litter loads are positively associated with the mismanaged plastic debris created in the river catchments [28]. However, this connection appears to be nonlinear in locations where large river catchments with high population density delivering an unduly higher portion of waste into the sea. The rivers of Yangtze, Yellow, Indus, Hai, Pearl, Ganges, Amur, Niger, Mekong and the Nile transport over 90% of global marine plastic waste into the ocean which originates from land-based sources [28]. In contrast, most of the debris on beaches away from urban centers (e.g., Alaska), is made up of fishing gear litter [40].

The tourism industry is both a significant contributor to ocean plastic debris and also a leading cause of the problem [8]. The presence of ocean plastic debris can demoralize some beach-loving visitors; thus, decreasing visitor attendances, which in turn can cause jobs and revenues losses in the tourism sector [41]. The potential effects of marine pollution identified in this contribution come into even more focus if viewed in terms of the well-being of future generations vis a vis their need for healthy seafood [42,43]. The discussion thus far has focused on identification of sources of marine plastic litter, the increasing evidence of the ubiquity of plastic litter in the open ocean, in the terrestrial environments and on shorelines of even the most remote islands [23]. Considering the sluggish growth in plastic recycling rates compared with the astronomical growth in plastic production [3], the volume of plastic litter in the marine environment has surely increased with time.

### 3.1.3. Potential Interventions

This section synthesizes peer-reviewed papers based on proffered interventions for ameliorating the volumes of marine plastic pollution. The review identified some critical studies, between 2010 and 2019, that suggested possible interventions to aid the reduction of MPP in the ecosystems. A significant number of the studies focus on freshwater (i.e., in-land waterbodies) while others focused on oceans (i.e., outland waterbodies). Most of the papers reviewed studied macro or microplastics, however, a few studied both [44–46]. Stringent regulation of plastic litter [47], surveillance [48], mitigation, prevention and collection of plastics dominated the proposed interventions by the studies reviewed. Other interventions focused on filtering of factory effluents and recycling of plastics. We summarized in Table 2 key studies possible intervention to reduce MPP. Table 2 presents details of the stated interventions.

**Table 2.** Freshwater.

| Summary of Key Marine Plastic Pollution Studies | | | | |
|---|---|---|---|---|
| **Types of Plastic Input** | **Environment** | **Location** | **Intervention** | **Reference** |
| Macroplastic | Freshwater | Seine River surface | Stringent regulation on plastic litter | [47] |
| Microplastic | Freshwater | Rhine River | Monitoring and reducing microplastic at point sources | [49] |
| Microplastic | Freshwater | Italy Adriatic Sea | Reduce plastic litter | [50] |
| Microplastic | Freshwater | Tibet river system | Increase collection capacity to reduce mismanaged waste | [51] |
| Micro- and macroplastic | Freshwater | Tamar estuary | Reduce ocean plastic pollution from land-based sources | [45] |
| Microplastic | Freshwater | Italy Lake Garda | Surveillance to control microplastic pollution in freshwater | [48] |
| Microplastic | Freshwater | China Lake Taihu | Prevent and eliminate pellet spillage | [52] |
| Microplastic | Freshwater | China, three Gorges Reservoire | Safely recycled and proper disposal of waste | [51] |
| Macro- and microplastic | Freshwater | Lake Canada beach | Increase filtering factory effluent | [46] |
| Microplastic | Freshwater | Rhine surface water | Reduce anthropogenic litter in Rhine River | [53] |

**Table 2.** *Cont.*

| Summary of Key Marine Plastic Pollution Studies | | | | |
|---|---|---|---|---|
| **Types of Plastic Input** | **Environment** | **Location** | **Intervention** | **Reference** |
| **Ocean: Summary of Key Marine Plastic Pollution Studies** | | | | |
| Macro- and microplastic | Ocean | South China Sea | Creating marine litter awareness through education | [44] |
| Microplastic, microbeads | Ocean | Deep-sea Ireland | Reduce microplastic emissions at source points | [54] |
| Macro- and microplastic, hard plastics | Ocean | Ocean, North Pacific subtropical gyre | Monitoring and tracking ocean plastic movements | [55] |
| Microplastic | Ocean | China beach | Regulating tourism activity | [56] |
| Plastic | Ocean | Kauai, Hawaii Beach | Substitute long degradable plastic with easily compostable ones | [57] |
| Microplastic | Ocean | Singapore seabed | Stopping pollution from vessels | [58] |
| Macro- and microplastic | Ocean | Mediterranean Sea | Increase monitoring actions to assess plastic pollution | [59] |

### 3.2. Theoreticl Studies

MPP models have been in existence since the 1970s and 1980s [22,60]. These models are useful for investigating the sources and impacts of MPP. Here we describe two of the most widely accepted models for understanding the distribution of MPP. One major theoretical model that has dominated the field of chemical pollution in recent years is the standard planetary boundaries model described in [61] and later applied to MPP by [6] to examine the impact of MPP on a global scale. Possible mechanisms and pathways for thresholds and global systemic change were identified [6]. Figure 3 shows a model that plots the relationship between plastic concentration and its effect on the ocean.

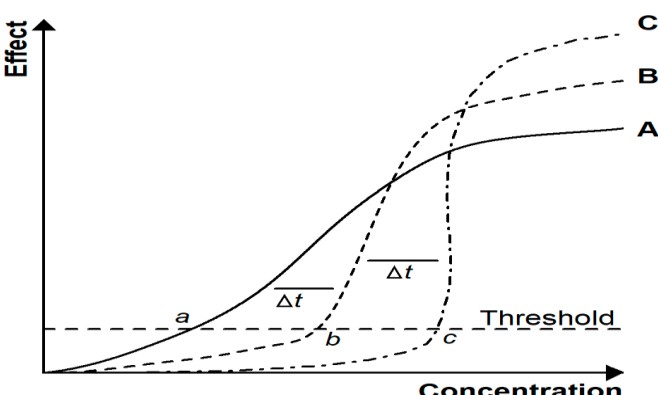

**Figure 3.** Cross-scale spatial and temporal dynamics of marine plastic pollution. Curve A depicts the direct local impact of plastic concentration on marine organisms which is being observed globally. Threshold (a) is the highest tolerable exposure of organisms. Curve B captures the impact on ecosystem; a large-region, time-lagged (Δt) impacts. Threshold (b) represents a basic ecosystem-level change due to the concentration of plastic in the ocean (e.g., loss of organism that has a major influence on the way the ecosystem works). Curve C denotes the cascading effect of plastic pollution on global ecosystem. A minimal impact of plastic pollution is noticed until a critical ecosystem shift occurs at the threshold c. [Adapted from [6]].

MPP appears to have direct impacts on organisms, indirect impacts as a vector (i.e., plastic litter is often seen as a potential vector for pathogens and viruses especially the deadly Zika virus) or bearer of other pollutants and systemic impacts that spread across ecosystems on multiple temporal and spatial levels [8] as shown in (Figure 3). However, four potential sequences of events by which the disruptive impacts of a given pollutant are not detected [62] until it turns into a global-level problem:

(i)  the concentrations of the pollutant are nearly homogeneous at a global scale;

(ii)    the impacts are swiftly distributed globally;

(iii)   the impacts of the pollutant are only detectable at a global scale; and

(iv)   when there is a time lag between the exposure of the pollutant and the impacts.

A planetary-level shift is a combination of many local changes; and plastic waste does not fit the boundaries model, because there was no plastic during the pre-industrial era [63]. However, there is debate about what forms a global-scale shift among the numerous frameworks of the planetary boundaries [63]. The Mediterranean Sea contained a relatively even distribution of plastic litter but with a higher density on the continental level with deleterious consequences for its inhabitants [64]. Plastic litter is being redistributed around oceans from dense concentration input regions which appears to fulfils MacLeod et al. [62]'s Scenario (i) demands [6] outlined above.

Whether MPP satisfies scenarios or sequence of events (ii)–(iv) depends on the impacts in question and on how 'planetary-scale' impacts are defined. The direct impacts of MPP on organisms (e.g., from entanglement, inhalation, ingestion) are often treated as isolated challenges–but they are gaining global attention because of their high accumulated levels as shown by curve A in Figure 3. As for Scenario (iv), the interval between the arrival of plastic into the oceans and its impacts on organisms is typically less than a decade [65]. However, considering ocean dynamics and the global redistribution of plastics, may contribute to the creation of high plastic hotspots in the marine environment which are geographically dispersed [13]. This framework is a useful way not only to show evidence that MPP meets planetary boundary threat conditions but also to identify knowledge gaps on boundary control variables that have not been rigorously studied. For instance, further transdisciplinary study is required to identify the thresholds of greatest concern in each stage of the planetary threat pathway.

Determining exactly how long plastic litter has been in the sea and their sources can be difficult but models are particularly useful for indicating possible transport channels and the period taken from the origin to sampling destination [66]. The transport and distribution of floating litters washed into the North Pacific Ocean as a result of the Tohoku tsunami in Japan was examined and suggests that a substantial quantity of litter washed into the ocean likely represents several years' worth of 'normal' debris flux from the country's urbanized coastline [60]. Predictions show the channels and impacts of vast amounts of plastic litters introduced via natural events such as tsunamis. Consequently, results also reveal a single event may drastically increase local plastic debris concentrations [60]. Although limited information is available for other sources of plastic into the ocean (e.g., input from natural disasters), the exact amount of plastic pollution caused by natural disaster are unknown.

Another best-known model is the Driver-Pressure-State-Impact-Response (DPSIR) first developed by the Organization for Economic Co-operation and Development [67]. The DPSIR framework was used to investigate dynamics of microplastic flows in the environment and identified population growth, economic growth after World War 2 and search for non-natural resources as key drivers [14]. It appears that environmental cleanups, environmental education campaigns, product design, the development of biodegradable plastics, waste management, the regulatory and policy instruments are potential responses to mitigate plastic pollution [14]. Table 3 presents an overview of the DPSIR model, which is a valuable adaptive management device for identifying solutions to environmental challenges [68]. The capacity of indigenous knowledge must be enhanced in order to overcome MPP. As outlined in Table 3, encouraging active participation of community-based organization and multidisciplinary experts can contribute to minimizing MPP [8].

**Table 3.** Major global marine debris challenges structured according to the Driver-Pressure-State- Impact-Response (DPSIR) approach. Source: Adapted from [68].

| Driver | Pressure | State | Impact | Response | Main Geographical Area |
|---|---|---|---|---|---|
| **Plastics Emitted to the Ocean from Coastal Areas** | | | | | |
| Use of plastic, especially in coastal zones. | 4.8–12.7 million tons of land-based plastic debris enter the ocean annually [4]. | The exact quantity of plastics in ocean unspecified; Some studies [8]; >100 million litters in 12 regional Seas [11]; possibly 51 trillion debris floating on the surface of the ocean [66]; South Pacific gyre average mass about 27,000 pieces per km$^2$ and 71 g km$^2$ [22]. | The environmental impact on marine ecosystems such as starvation of marine species or entanglement or; economic impacts on tourism through blocked waterways or littered shores [8]. | Short-term solutions: A portfolio of diverse solutions such as source reduction, innovations in ocean cleanups technologies, improve waste management infrastructure, [69]; Long-term solution: adopt circular economy thinking -through behavioral change; reduce single use plastic; phase out non-recoverable plastics; encourage alternative materials [3]. | Worldwide short-term focus on Asia. Especially middle- and low-income economies (top 5: China, Philippines, Indonesia, Sri Lanka, Vietnam); Beaches clean up; Ocean Clean ups: North and South Pacific gyres; 'hotspots' of plastics. |
| **Macroplastics Emitted from Rivers** | | | | | |
| Use of plastics, especially in river catchment. | 1.15–2.41 million tons of plastic debris flows from rivers as a result of mismanaged debris or population growth [29]. | The exact quantity of plastics in ocean undocumented; Some studies [8]; >100 million debris items in 12 Regional Seas [11]; perhaps 51 trillion debris on ocean surface [66]. | The environmental impact of plastic on marine ecosystems such greenhouse gas emissions [2]; ocean damage [70]. | Short-term solution: Indigenous knowledge and the capacity of the local communities should be encouraged to address plastic pollution [71]. Long-term solution: Enhancing wastewater treatment facilities in developing countries. Encourage more scholarly interdisciplinary research into causes, effects and responses to contributes to reducing plastic waste in marine environment [72]. | Worldwide but short-term focus on Asia; over 60% of the most polluted rivers are based in Asia [29]. |
| **Macroplastics from Abandoned Lost or Otherwise Discarded Fishing Gear** | | | | | |
| Fishing. | The exact emission unknown; About 640,000 tons per year [8]. | Approximately 10% of world marine litter by volume [34]. | Habitat damage and entanglement of marine mammals via ghost fishing [26] and impacting habitats of conservation [73]. | Short-term Solutions: legislation ALDFG now only aiming at large scale vessels (>100 gross tonnage) should also aim at smaller ships. paying fishers to remove ALDFG or marine debris for recycling [74]. Long-term solutions: Encouraging preventative approaches and quick recovery of ALDFG [75]. | Global. |
| **Primary Microplastics** | | | | | |
| Use of microplastics; production flakes or microbeads used in industrial abrasives both on land and at sea [76]. | Existing emission levels not yet known. | About 32,000–236,000 metric tons microplastics in oceans [76]. | Perhaps ecotoxicological impacts, economic damage because of food safety concerns and accumulation in food chains. | Short-term solutions: prevention of microplastics entering the ocean; improve wastewater treatment facilities; bans like microbeads; industrial spills. Long-term solutions: as outlined above and improved technologies and alternative materials. | Global. |
| **Secondary Microplastics** | | | | | |
| Crumbling and disintegration of plastics; tear and wear of tires; disintegrated packaging [76]. | Existing emission levels not yet known. | About 32,000–236,000 metric tons microplastics in oceans [66]. | Perhaps ecotoxicological impacts, economic damage because of food safety concerns and accumulation in food chains. | Short-term solutions: prevention of single use plastics; upgrade waste water treatment facilities. Long-term solutions: Improve technologies such as filters washing machines. | Global. |

Overall, there seems to be some evidence to indicate that MPP fulfils circumstances for chemical contamination of planetary boundary. In particular, there is widespread evidence of the marine ecosystem consequence of plastic pollution but it is debatable whether MPP fulfils the overarching condition of disrupting earth system dynamics [62]. A consideration of the DPSIR framework for MPP, however, highlights the possibility of using the *Response* element of DPSIR framework to reduce the loss of marine ecosystems by influencing behavior change and utilizing market-based instruments such as bans [68,77]. The introduction of deposit refund schemes on plastics and other economic incentives encourage recycling and stimulate behavior that alleviates the marine litter problem as well as imposing higher taxes on shops and supermarkets using a lot of non-recyclable package [78].

## 4. Conclusions

The purpose of the current study was to examine both theoretical and empirical overview of marine plastic pollution based on existing studies and suggest future recommendations. Empirical studies outlined the sources and prevalence of plastic litter in the marine ecosystem while theoretical studies clarify under what conditions MPP becomes a large-scale problem. Though empirical investigations are limited due to lack of data on the number of plastic wastes entering the oceans [13], empirical studies provide systematic evidence on the impact of MPP in the real world. Recently, there has been an increasing amount of robust scientific data (e.g., neutrally-buoyant floats, current meter arrays, satellite surveillances and oceanographic observations of salinity and temperature) on MPP. Hence, there is increasing scope for more studies. Notably, a recent detailed assessment of the field and laboratory-based observations of plastic litter and fragments on a broad variety of marine organisms such as ingestions have been conducted by the Joint Group of Experts on Scientific Aspects of Marine Environmental Protection (GESAMP) [76]. This assessment outlines potential areas for future research. For instance, detailed field and laboratory information indicate the negative impacts of exposing marine organisms to marine plastic pollution [76,79]. For example, the assessment includes the impacts of ingestion and retention of microplastic by Blue mussels (Mytilus edulis) [76]. In addition, a detailed compilation of results for both commercial and non-commercial fish species were explored [80]. As such, the scope of the assessment has expanded, nevertheless, it does not yet include long term human health impacts.

Though it seems that river catchments, particularly those with high coastal populations and high levels of coastal tourism, can transport a significant load of plastic into the marine ecosystem as outlined in the GESAMP assessment [76], detailed information on the quantity and types of plastic litter getting into the marine environment globally, which causes are most significant and what control measures may be most effective are lacking. Given the scope of GESAMP assessment, large investments into global monitoring are warranted to produce the datasets necessary for regional and global comparisons. In highlighting the regional diversity in the sources, distribution and effects of marine plastic litter, we argue that reduction measures and policies must take these variations into consideration.

Another potential area for future empirical work is to assess the degradation rate of various plastic resins in the marine ecosystem and the period of potential exposure to marine organisms. As marine plastic debris is not distributed randomly across all oceans, it is demanding to create a control group of marine organisms, for instance, commercial fishers that can be utilized to examine a counter-factual studies. In this regard, water columns need to be stratified to obtain sampled population that can be used to explain the behavior of plastic litter inside the different layers of the ocean.

To minimize the plastic footprint on the marine ecosystem, innovation, monitoring and regulations are critical but to do this requires enhancing opportunities for the exchange of ideas between researchers and policymakers. All of these require adequate financing [81]. Participation of economists and other academic experts on conferences related to MPP such as the United Nations Conference on Sustainable Development, entitled: "The Future We Want" is limited to a handful of representatives mostly from developed countries. Most attendees at International conferences on plastic in the marine environment (ICPME) are diplomats and environmentalists who are not privy to recent academic

studies that enhance our understanding of MPP. Here, we identify the need to develop a delegation of representatives from Small Island Developing States and least developed economies to strengthen the links between scientific knowledge and the practical implications of MPP across countries and contexts worldwide.

More importantly, citizens must take individual actions not only to help reverse the trend of MPP by demanding integration of recycled content in plastic products but also take ownership of the monitoring and implementation of SDG 3 as well as SDG 12 and SDG 14. Major gains towards SDG 3 will be achieved through the introduction of regulatory mandates on certain polymers, pigments and additives, via a global plastic protocol to prevent health risk and increase plastic recyclability. Controlling virgin plastic production and consumption particularly of single-use, low value, disposable plastics will lead to the realization of SDG 12, while SDG 14 can be achieved via the effective management of plastic from land-based sources by providing downstream infrastructure such as good waste collection systems, especially, for high pollution countries.

Ocean circulation models can contribute useful assessments of the distribution and relative prevalence of floating plastics; however, global-scale debris modelling may underestimate in some areas while over-estimating in others [13]. In addition, the main drawbacks with many current global-level modelling techniques are that they do not examine key factors such as non-buoyant plastics, vertical transport towards the seafloor and fragmentations [8]. The empirical analysis of plastic litter in the ocean suggests that surface plastic debris levels in gyres are lower than model debris predictions would suggest [30]. A reasonable approach to tackle this issue requires sound metrics to prioritize action at regional and local levels, ranging from proper infrastructure to innovative product design for plastic recycling that cover many types of plastic polymers.

**Author Contributions:** Conceptualization, writing the original draft preparation, and review I.I.; resources, editing and supervision U.R.S. Both authors have read and agreed to the published version of the manuscript.

**Funding:** This research received no external funding.

**Acknowledgments:** Support for this project was provided by The Pew Charitable Trusts. The views expressed are those of the authors and do not necessarily reflect the views of The Pew Charitable Trusts.

**Conflicts of Interest:** The authors declare no conflict of interest.

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
