# Peer review of "A Review of the Production, Recycling and Management of Marine Plastic Pollution"

_jmse, doi:10.3390/jmse8110945_

Round 1

Reviewer 1 Report

There is no mention in the introduction regarding the aim of the paper so that the evaluators are able to understand the remaining text.

I cannot find any actual economic figure in the whole text. Most information presented is actually environmental and not economic (mostly amounts and some rates)

Author Response

We appreciate the useful comments and suggestions. kindly find attached our responses

Reviewer 2 Report

The topic of this manuscript fits well with the journal. However, the content of this manuscript is not well organized and it is also not well formatted. Major revision is suggested for now.

  1. The difference between the descriptive studies and empirical studies is not quite clear in the manuscript. In the descriptive studies, the authors review studies which quantify the plastic debris in the ocean or coastal areas, and also their potential impacts to the living creatures. These estimates of plastic debris are not descriptive at all, instead they are quantitative or semi-quantitative studies. In the empirical studies, the authors review the specific ways how the plastics in the ocean could cause problems to the living creatures. They are more like descriptive research or case studies. Overall, the authors may reconsider the way they categorize these marine plastic studies.
  2. All figures need figure caption.
  3. Figure 1 does not help much in the content. It could be removed or put Figure 1 after Figure 2, because figure 1 is only cited at line 183, which is after the citation of figure 2.
  4. Line 194, a typo. Capitalize ‘H’ in however.
  5. Line 203, a typo. It should be curve A in Figure 2.
  6. Line 381-424. Need to edit or remove some of these texts.

Author Response

Thanks for the insightful comments and suggestions. Kindly find attached our responses.

Reviewer 3 Report

This paper reviews recent studies on marine plastic pollution (MPP) and provides suggestions for future research. The authors categorized previous studies into three categories: descriptive, theoretical, and empirical. The descriptive studies include the ones defying MPP, identifying sources and distribution of plastic litter, and estimating the amount/rate of MPP. The theoretical studies provide models to explain or predict the sources and impacts of MPP, and the empirical studies provide empirical data of the sources and impacts of MPP.

This review is comprehensive and informative. I am sure it will be beneficial to the others studying marine plastic/marine litter pollution. Here I am listing the questions I have when reading the manuscript, and hope the authors can clarify them in their revised version:

  1. The title “A review of economic studies” implies that this paper will focus on the economic perspectives of the impacts of marine plastic pollution. However, I found it is less clear in the presented review. Only a handful of studies directly discuss the economic impacts of MPP. I would suggest the authors explain more about the inclusion criteria for studies reviewed in this manuscript.
  2. The difference between “descriptive studies” and “empirical studies” is not clear enough. I was confused with why some studies were classified as “descriptive” studies but not “empirical” studies, while they provide empirical observations and estimates of MPP. Jambeck et al. (2015) was even reviewed in both sections.
  3. The possible intervention to reduce MPP (Table 3) was presented in the “empirical studies” section, which confused me again. I agree that this is a very important information to be included. However, it will be better to have a new section called “Potential interventions” and to provide more detailed discussions on these studies.
  4. Line 164: “…Here we describe two of the most widely accepted models…” After you introduced the first model (standard planetary boundaries model), I was confused about which model is the second model you intended to describe.

Round 2

Reviewer 2 Report

Although the authors have made a lot of improvement, the figure captions they edited are too simple. It needs to be clear that figure caption is different from a figure title. For example, in figure 1, the authors need to provide what does 3, 14, 12 mean, or why the authors use those three numbers instead of 1, 2, 3. In figure 2, what does the number inside the parenthesis indicates after each species, and what does the number after each blue bar mean. Same problem in figure 3. And these captions must be edited before the manuscript is accepted.

Author Response

This manuscript is a resubmission of an earlier submission. The following is a list of the peer review reports and author responses from that submission.

Round 1

Reviewer 1 Report

Dear authors,

please find in the attached file my comments

Reviewer 2 Report

The paper “Marine plastic pollution: A review of economic studies” by Issifu and Sumaila is an attempt of synthesizing the actual knowledge about this important issue, but there are several weaknesses in this paper that lead me to recommend the rejection of this article given that it has serious flaws.

The title is confusing: The title invites to expect a revision of economic studies, but economy is not the core of the revision at all. This is obvious when the authors state their ambitious aim in page 2: “the purpose of this paper is to present a theoretical and empirical overview of marine plastic pollution, and suggest future recommendations in accordance with the lessons learnt from the current study”; nothing about economy.

The article does not contain new information, it has no own results nor methods, it is a bibliographic review of existing literature, although the revision is lacking important articles that should be considered. This is also the case of table 3, announced as a summary of key marine plastic pollution studies, but it is lacking a lot of important studies.

There is an inappropriate use of figures and tables. Most of them (i.e. fig 2, fig. 3, table 2) are a literal copy of the graph or table in another article. For example figure 2 in this paper is the exact copy of figure 2 in the paper by Villarrubia-Gómez et al. (2018), while in the figure legend nothing is said about this plagiarism. A similar total copy can be seen in table 2, which is a literal copy of a table within the paper by Löhder et al., 2017. Figure 1 is unnecessary, the message can be changed by two lines of text, without losing any information. Indeed, there is no reference to this figure 1 in the text. Moreover, there are erroneous references to figure 1 (for example at pages 8 and 10) when authors try to refer to figure 2. Table 3 appears in the article previous to tables 1 and 2…

Finally, the carelessness with the figures and tables is similar to the one related with arguments and ideas. The paper is a mixture of known arguments without proper organization, with figures and graphs taken from other authors without permission nor recognition, and without a economic analysis as the title promised.